# Copper, Zinc, and Selenium Levels During Pregnancy and Their Relationship with Pre-Eclampsia Risk: A Systematic Review

**DOI:** 10.3390/nu17061038

**Published:** 2025-03-15

**Authors:** Pietra dos Santos Goularte, Carlos Henrique Lima Imperador, Felipe Hassan Piedade, Amanda Nascimento Guerin, Karina de Toledo Martins, Diogo Boreski, Edmo Atique Gabriel, Sthefano Atique Gabriel, Daniel Guimarães Tiezzi, Chung Man Chin

**Affiliations:** 1Advanced Research Center in Medicine (CEPAM), School of Medicine, Union of the Colleges of the Great Lakes (UNILAGO), Sao Jose do Rio Preto 15030-070, SP, Brazil; pietragoularte24@gmail.com (P.d.S.G.); carloshlimperador@gmail.com (C.H.L.I.); felipe.hassan@hotmail.com (F.H.P.); amandaguerin@terra.com.br (A.N.G.); karina.toledo.martins@gmail.com (K.d.T.M.); edag@uol.com.br (E.A.G.); sthefano@unilago.edu.br (S.A.G.); dtiezzi@usp.br (D.G.T.); 2Laboratory for Drug Design (LAPDESF), School of Pharmaceutical Sciences, University of São Paulo State (UNESP), Araraquara 14800-903, SP, Brazil; diogo.boreski@unesp.br; 3Gynecology Department, School of Medicine of Ribeirão Preto, University of São Paulo (USP), Ribeirão Preto 05508-220, SP, Brazil

**Keywords:** pre-eclampsia, pregnant, micronutrient, copper, selenium, zinc

## Abstract

Pre-eclampsia (PE) affects about 5% of all pregnancies worldwide and is one of the leading causes of maternal and fetal morbidity and mortality. Some reports suggest that micronutrients may influence this condition, but there is no existing report analyzing data on copper (Cu), selenium (Se), and zinc (Zn) regarding quality and bias. Accurate information is crucial to support governmental healthcare actions, especially in developing countries such as Brazil. This study aims to investigate whether alterations in Zn, Cu, and Se levels in pregnant women contribute to the development of PE. We conducted the study following PRISMA guidelines and registered it on PROSPERO (CRD42022302298). We searched LILACS, PubMed/MEDLINE, EMBASE, and the Cochrane Library databases from January 2000 to January 2024. Of 1202 reports, 42 manuscripts were suitable for analysis (contained one or more micronutrients). A total of 76 individual analyses (by nutrient) were performed using Joanna Briggs Critical Appraisal and Cochrane Risk of Bias Tools. The analyses classified 69 reports as fair/low-quality with bias. Due to this, a meta-analysis was not conducted, as the results would not have accurately reflected reality. Of the high-quality reports, five on Cu showed conflicting results, while two studies on Zn found no differences in the level of this micronutrient between normal and PE pregnancies. No high-quality studies were identified for Se. The results highlight the need for robust guidelines for research involving micronutrients and PE to address this question effectively.

## 1. Introduction

Micronutrient deficiencies are of great importance in Public Health when considering women of reproductive age in low and middle-income countries (PBMR) [1,2,3,4]. According to the International Federation of Gynecology and Obstetrics (FIGO), globally, 76,000 women and 500,000 babies die each year from pre-eclampsia (PE), and women in low-income countries are at a higher risk of developing PE compared to those in high-income countries [5]. The most common cause of death in women with PE is intracranial hemorrhage; other serious complications include placental abruption, HELLP (Hemolysis, Elevated Liver enzyme, Low Platelet) syndrome, acute pulmonary edema, respiratory distress syndrome, and acute renal failure. PE is also associated with several short and long-term perinatal and neonatal complications, including death [5].

According to Roberts et al. [6], the definition of PE has evolved significantly, affecting its diagnosis and epidemiological understanding. The guidelines from health organizations no longer require proteinuria for diagnosis, instead focusing on new onset hypertension (≥140/90 mm Hg) with other systemic symptoms. This shift complicates comparisons with older data and highlights the complexity of PE as more than just high blood pressure. There are ongoing discussions about lowering the diagnostic threshold to ≥130/80 mm Hg, balancing the potential benefits with increased monitoring needs.

Currently, PE is classified by the timing of diagnosis: early-onset, linked to poor placentation, and late-onset, related to a mismatch between maternal blood flow and fetal needs, as well as a predisposition to cardiovascular issues. Understanding the underlying causes of vascular dysfunction in PE is crucial for developing better prediction and prevention strategies. The syndrome has multiple causes, leading to endothelial dysfunction, inflammation, and stress on the placenta. The recent link between COVID-19 and PE raises questions about whether it is strictly a pregnancy-related condition. More research is needed to clarify the contributions of several factors to the disorder [7].

Inflammatory processes can lead to metabolic dysfunctions and nutrient losses, which, during the pre-conception, post-conception, and pregnancy periods, may result in fetal alterations, such as neurological disorders in newborns caused by folate deficiency. To address this issue, folate supplementation has been included in pregnancy prescriptions [8]. Regarding other nutrients, such as vitamin B6, their deficiency has been suggested as a potential risk factor for PE [9]. However, a systematic review conducted by Salam et al. [3] that included four clinical trials identified significant limitations, such as high risk of bias, unclear trial methodologies, and inadequate follow-up. As a result, the review did not provide sufficient evidence to support the clinical benefits of vitamin B6 supplementation during pregnancy.

Micronutrients such as calcium, vitamin D, vitamin E, vitamin C, and iodine and their relationship with PE have been extensively studied [10,11,12], allowing for a safer decision on the prescription of these micronutrients as supplements during pregnancy.

Vitamin D has been associated with potential reduction in the risk of PE [13,14]. A comprehensive network meta-analysis published in 2022, which included data from 130 randomized controlled trials (RCTs) involving 1,122,916 participants, identified several effective interventions for PE prophylaxis. These interventions included low-molecular-weight heparin (RR: 0.60, 0.42–0.87), aspirin (RR: 0.79, 0.72–0.86), vitamin D (RR: 0.65, 0.45–0.95), calcium (RR: 0.71, 0.62–0.82), and exercise (RR: 0.68, 0.50–0.92) [11]. However, more recent evidence has questioned the effectiveness of vitamin D supplementation in preventing PE. Three systematic reviews, including a Cochrane review published in 2024, found no clear benefit of vitamin D in reducing PE risk. Although vitamin D has been considered a potential preventive measure, the current findings remain inconclusive, and its role in PE prevention is still uncertain [13,14].

Most of the micronutrients, including copper (Cu), selenium (Se), and zinc (Zn), lack enough studies to allow their real necessity of prescription to prevent the development of PE.

Zn is a trace element, ranking as the second most abundant metal in the human body after iron (Fe). It serves important structural, regulatory, and catalytic roles and is essential to over 2500 proteins, including various enzymes and transcription factors [15,16]. Approximately 10% of the human proteome depends on Zn ions, which play a significant role in gene regulation, DNA metabolism, chromatin organization, cell growth, programmed cell death, immune response, cognitive function, and antioxidant defenses [15]. Additionally, Zn is crucial for intracellular signaling and essential for proper synaptic activity in the central nervous system [17].

Cu-Zn superoxide dismutase (Cu/Zn-SOD) is a metalloenzyme responsible for converting the superoxide anion (O₂⁻) into molecular oxygen and hydrogen peroxide (H₂O₂) 18]. Reactive oxygen species (ROS), such as superoxide anions, are unstable and can damage DNA, potentially disrupting the DNA damage response (DDR)—a critical issue in cancer, where DDR pathway mutations are common [18]. Recent findings indicate that Cu/Zn-SOD also influences the expression of genes responsive to ROS, including certain oncogenes [19]. Furthermore, Cu/Zn-SOD appears to protect against DNA strand breaks caused by superoxide, underscoring its potential significance in cancer treatment [20]. Because Cu/Zn-SOD relies on both Cu and Zn for functionality, the balance of these metals in the blood, rather than their absolute levels, is essential for DNA stability.

Se is present in plants and animal tissues, and its concentration depends on geographic location and soil region [21]. Its deficiency in the human body can affect the cardiac, reproductive, and immune systems [22]. It is an important trace element that integrates selenoproteins such as glutathione peroxidase (GPx), iodothyronine deiodinases, thioredoxin reductase, and selenoprotein S, P, and W found in the human placenta [22,23]. Selenoproteins contain selenocysteine in their active site and act to protect the endothelium, as they reduce inflammation by controlling the expression of the adhesion molecule, which is induced by cytokines [24,25].

Glutathione peroxidase, thioredoxin reductases, selenoprotein P, and superoxide dismutase (SOD) participate in the natural antioxidant defense system within the human placenta [26,27]. During pregnancy, Se levels and glutathione peroxidase activity typically decline. To enhance antioxidant activity, a daily intake of 60 µg of Se is recommended [21,23]. The explanation for the etiopathogenesis of PE is still unknown; however, some studies suggest that pre-eclampsia may be linked to the exacerbation of inflammation, which can cause problems in the endothelium [26]. Additionally, other studies indicate that supplementation with Se and other antioxidants could reduce the risk of developing PE [28,29].

The present study aims to perform a systematic review according to PRISMA guidelines, using the Cochrane risk of bias tool and the quality of the report and Joanna Briggs Critical Appraisal Tools to determine whether Zn, Cu, and Se deficiency in pregnant women might develop or induce PE to support clinical and governmental healthcare actions.

## 2. Materials and Methods

We searched for clinical studies published between January 2000 and January 2024 in PubMed/Medline, Embase, Lilacs, Cochrane Library, and ClinicalTrials.gov databases using Medical Subject Heading (MESH) terms for micronutrients, PE, eclampsia, Zn, Cu, Se, woman, and pregnant. The review was performed according to PRISMA guidelines and was registered in PROSPERO (CRD42022302298).

Inclusion Criteria: population: pregnant women compared to controls (non-pregnant); micronutrient intervention; blood, urine, and/or tissue micronutrients (Cu, Se, and/or Zn) determination. Types of studies eligible for inclusion: randomized trials, case–controls, cohort studies, and case reports written in English or Portuguese.

Exclusion Criteria: population: animal research, non-pregnant women, lack of micronutrient measurement (Cu, Se, Zn); micronutrients analyzed after delivery. Non-eligible studies: reviews, comments, abstracts, points of view, posters, and editorials.

Data Extraction: Two independent groups with four researchers each (PG, AG, FHP, DGT) (CHI, KTM, DB, SAG) selected the reports after analyzing the titles and abstracts. After comparing the results, duplicate reports were excluded. The selected reports were analyzed by the two independent groups and data were extracted. Disagreements were solved through debate and if necessary, by a third set of researchers (CMC, EAG). Missing data was obtained by contacting the authors via email.

Risk of bias and quality assessment: The risk of bias characteristics was assessed, including method of randomization, treatment allocation, micronutrient analysis, blood arterial pressure measurement, age, sex, time of sample collection, period of gestation, and absence of comorbidity information, including obesity (Body Mass Index—BMI), diabetes, previous hypertension, and renal diseases, which decreased the quality of evaluation. We used the Cochrane risk of bias tool to classify the studies as green (no/low), yellow (fair), and red (high). The quality of the reports was analyzed using Joanna Briggs Critical Appraisal Tools. After analysis, the reports were classified as green (good), yellow (fair), and red (bad). Primary outcome: PE; Secondary outcome: hypertension without proteinuria.

## 3. Results and Discussion

Women with PE are four times more likely to develop chronic hypertension and have approximately twice the risk of coronary artery disease, stroke, and thromboembolism [30]. This review seeks to support medical and governmental clinical actions to address these risks. Maintaining adequate nutritional status during a woman’s reproductive years is essential for preparing the body for a healthy pregnancy and fetal development, particularly during the first trimester, the most critical period for the development of various organs [31]. This review focuses on the role of Zn, Cu, and Se in the context of PE.

The flowchart below (Figure 1) shows how our selections were made and the respective amounts of each micronutrient. The flowchart and tables show the reports by individual micronutrients (Cu, Zn, Se) found.

The discrepancy between the total number of articles analyzed and the number included or excluded arises because some reports cover multiple micronutrients and are, therefore, counted more than once in the flowchart: an article could be excluded for one micronutrient but still be relevant for another, leading to its inclusion in certain analyses. Each micronutrient in a multi-nutrient study is independently evaluated and documented in the corresponding table for that nutrient. For example, a study involving Zn, Cu, and Se would have data for each micronutrient listed separately in their respective tables. In cases where quality classification discrepancies occurred, a third group of reviewers conducted a re-analysis to reach a final decision.

The 76 reports in the flowchart correspond to 42 manuscripts suitable for analysis (contained one or more micronutrients). The 42 manuscripts analyzed are shown in Table 1.

The analysis was performed individually (by micronutrient), with 69 classified as low-quality reports with bias. The following tables present the individual evaluations conducted for each paper. The papers were cross-analyzed and final scores were assigned, with the lowest score chosen as a precautionary measure. Table 2 summarizes the findings for the micronutrient Cu, Table 3 for Zn, and Table 4 for Se.

Table 3 presents the results of the analysis for the micronutrient Zn. Among the 27 reports evaluated, a significant proportion—14 studies (51.8%)—were classified as low quality (red). Only two reports (7.4%) were rated as high quality (green), one with no/low risk of bias and one with moderate risk, while 11 (40.7%) were considered of moderate quality (yellow). Additionally, 17 studies (62.9%) were found to have a high risk of bias (red), underscoring the considerable limitations in the reliability of these findings. Seven reports (29.6%) were identified as having a moderate risk of bias.

Table 4 outlines the results of the analysis of the micronutrient Se. Notably, none of the 18 reports were rated as high quality (green, 0%), indicating a complete absence of studies with no or low risk of bias. A significant majority—11 studies (61.1%)—were classified as low quality (red), reflecting substantial concerns regarding the overall standard of research in this field. The remaining seven reports (38.9%) were considered moderate quality (yellow). Furthermore, 12 studies (66.7%) demonstrated a high risk of bias (red), while six reports (33.3%) were identified as having a moderate risk of bias (yellow). The following tables were background colored to facilitate the readability for each micronutrient.

Analyzing an isolated micronutrient in pregnant women is challenging because most reports include multiple nutrients or other parameters, and the methods for quantifying micronutrients and the biological materials used vary (blood, placenta, umbilical cord, toenail). Our systematic review showed that most manuscripts did not adequately describe the identification of PE and failed to describe comorbidities, clinical conditions such as BMI and/or proteinuria, and age differences. Some studies also used men or non-pregnant women as controls, further diminishing the quality of the studies. In addition, the manuscripts did not show or discuss the normal clinical reference level of micronutrient range and the increase or decrease in micronutrient levels between groups, leading to bias in the conclusions.

Most reports on Cu did not analyze it alone; instead, they examined its association with other micronutrients such as Zn (because of the Cu/Zn-SOD relationship), Se, and magnesium (Mg), among others. Kiran et al. [46] suggested that one of the main causes of PE is oxidative imbalance and decreased antioxidant activities. Superoxide dismutase (SOD), an antioxidant enzyme preventing damage caused by free radicals, and Zn serve as enzymatic cofactors. Therefore, alterations in the homeostasis of this mineral can trigger oxidative/inflammatory processes such as PE. Additionally, pregnancy is associated with increased Cu retention, likely due to hormonal changes that reduce biliary Cu excretion [68]. The normal ranges of Cu, Zn, and Se (serum sample) according to Abbassi-Ghanabati et al. [69] are shown in Table 5.

Several studies have indicated a significantly lower concentration of serum Cu in patients with severe PE. However, most of these studies exhibit at least one significant bias or error in their design. For instance, Lewandoswka et al. [24] selected a participant group with an age range of 18 to 45 years but did not consider age as a predisposition to increased occurrence of hypertension. Another study by Enebe et al. [27] demonstrated lower Cu levels in the study group compared to the control, but statistical analysis revealed similar levels when considering the standard deviation stipulated by the study. Furthermore, this study did not provide information on gestational age or the study period, constituting a second bias that diminishes the study’s quality.

Of the reports involving Cu analysis, only five out of 31 were considered good quality and without or with low bias. For Zn analysis, one report out of 27 studies was considered good and there was no report considered good for Se from 17 analyses. Good quality reports for Cu, such as Fenzl et al. [45], Elmugabil et al. [47], and Gul et al. [56], concluded that there are no differences in Cu levels between normal pregnant or PE women, while Rafeeinia et al. [50] and Polat et al. [38] showed an increase in Cu levels in PE. Rafeeinia et al. [50] showed an increase in Cu levels in mild PE and severe PE compared to normal women in the third trimester of gestation (the values found for normal pregnancy were in accordance with reference values and for mild PE were similar to the second trimester (2.26 ± 0.48 μg/dL). For severe PE, they found 270 ± 85 μg/dL). Polat et al. [38], studying PE’s correlation with thrombophilia in Turkish women, found, as an independent marker, higher concentrations of Cu in PE women (210.95 ± 56.58 μg/dL) in the third trimester. The concentration in normal pregnant women at the same period (121.21 + 25.08 μg/dL) was elevated above the reference value, introducing potential bias in the study findings.

Ilhan et al. [49] did not provide details on the inclusion and exclusion criteria or demographic characteristics of the study participants, limiting the ability to generalize the findings. Although proteinuria was reported in 21 patients, it was observed in only 20 of the 22 patients included in the study, suggesting possible discrepancy in the data. The study also did not address BMI, a critical factor in the context of PE, which may impact the interpretation of the findings. Cu levels were within the normal range, despite minor variations, whereas Zn levels were consistently above the normal reference range. Notably, Zn levels were elevated in both normal and non-pregnant women, exceeding reference values and potentially introducing bias in the study outcomes. This elevation in Zn levels, particularly in non-pregnant women, may skew the interpretation of micronutrient status in relation to PE.

Atamer et al. [64] investigated serum levels of Cu, Zn, and Se, along with serum malondialdehyde (MDA), leptin, placental MDA, and glutathione (GSH). In the third trimester, Cu levels were elevated in PE patients compared to healthy pregnant women but remained within the reference range for normal pregnancy and non-pregnant groups. Regarding Zn levels in PE, the values found were consistent with references, whereas they were elevated in the normal pregnancy group. For Se, typical levels were observed in both non-pregnant and normal pregnancy groups, but a slight reduction was noted in PE patients. Although a significant negative correlation between Cu and Se levels was found in pre-eclamptic women (r = −0.407, *p* < 0.05), the authors did not contextualize this finding in relation to reference values or investigate potential associations with Cu/Zn superoxide dismutase (SOD) activity. Moreover, the study did not compare findings to established reference data or account for potential micronutrient supplementation during pregnancy, limiting the interpretability and broader implications of the results.

Kashanian et al. [39] supplemented Cu at a dosage of 1000 mg during pregnancy to assess its effects on pregnancy outcomes and the incidence of premature rupture of membranes. They did not observe any adverse pregnancy outcomes, though they noted a reduction in maternal depressive symptoms. However, the age range of the participants was broad (18–35 years), potentially introducing variability, and the study did not measure Cu levels before or after supplementation, limiting insight into how supplementation may have impacted serum Cu levels directly.

In contrast, Keshavarz et al. [52] and Jamal et al. [55] did not indicate in their exclusion criteria whether patients were using micronutrient supplements, which may have influenced their findings, as their reported Cu levels for normal pregnancies were notably higher than reference values. Specifically, Elmugabil et al. [47] reported Cu levels of 2.76 ± 1.2 mg/L (276 ± 120 μg/dL) and Abo-Elmatty et al. [53] reported levels of 0.39 ± 0.02 mg/dL (390 ± 20 μg/dL), both exceeding the typical reference range of 130–140 μg/dL. In PE patients, Keshavarz et al. [52] observed Cu levels of 2.01 ± 0.83 mg/L (201 ± 83 μg/dL),

Enebe et al. [27] did not specify the timing of sample collection for determining Cu concentrations, making it challenging to accurately compare their findings with reference data. Nevertheless, their results indicated that patients with PE had Cu levels like those of non-pregnant women, while normal pregnant women had Cu levels within reference limits for the second trimester. Sak et al. [40], in contrast, collected samples after the 20th week of gestation—corresponding to the second or third trimester—but did not discuss their findings in relation to reference data. In their study, normal pregnant women exhibited Cu levels of 62.6 μg/dL, which are below the standard reference range. This underestimation of normal pregnancy values could introduce bias in the interpretation, as it may exaggerate the apparent elevation of Cu levels in PE patients.

Açikgoz et al. [44], Kolusari et al. [66], Negi et al. [29], Katz et al. [63], and Rayman et al. [28] analyzed the micronutrient levels in samples from various tissues, including placenta, umbilical cord, and toenail, without reporting reference ranges. These samples were deemed less representative of gestational nutritional status, as micronutrient concentrations in the placenta and umbilical cord can fluctuate during labor due to physiological changes, placental function, maternal stress, umbilical blood flow variability, and hormonal shifts. Rayman et al. [28] uniquely analyzed toenail samples collected before delivery, between 3 and 12 months prior; however, the exact timing of collection was not specified, nor were comparisons made across gestational periods. Furthermore, there is no established reference range for Se in toenail tissue, complicating any reliable interpretation or conclusions from these findings.

Several studies, including those conducted by Ilhan et al. [49], Serdar et al. [36], Kiran et al. [46], and Monia et al. [41] compared Cu levels in PE women with those in normotensive pregnant women. However, their conclusions also included comparisons of Zn levels between PE and non-pregnant women, introducing significant bias due to the lack of consistency in the comparison groups.

In the study performed by Serdar et al. [36], no comorbidities were reported beyond hypertension. While Cu levels were measured through ceruloplasmin quantification—a potentially useful tool—its accuracy as an indicator of Cu status is limited. For a more reliable assessment, it is recommended to complement ceruloplasmin measurements with direct serum and urinary Cu quantification, as noted by Roberts and Schilsky et al. [70]. This approach would improve the precision of Cu status evaluations, providing a more robust basis for analyzing Cu-related outcomes in pregnancy.

In the report by Onyegbule et al. [51], patient stratification was insufficient, as both young and advanced maternal age are recognized as risk factors for PE. The study reported a mean age of 27 ± 7 years, with ages ranging from 15 to 37 years, yet failed to control age-related risks. Cu values for normotensive pregnant women were below the normal range, while those for PE patients fell within normal limits. Additionally, sample collection occurred only after the 20th week of gestation, which limits insight into Cu levels earlier in pregnancy.

The Farzin et al. [65] study included participants with a broad BMI range (19–30), thus combining non-obese and obese pregnant women, with mean BMI values of 26.84 ± 2.15 for normotensive and 27.11 ± 3.22 for hypertensive groups. However, no further clinical information was provided aside from blood pressure and hemoglobin levels, limiting the study’s depth in analyzing PE-related factors. In contrast, Al-Jameil et al. [20] reported contradictory inclusion and exclusion criteria by defining a BMI > 35 as an inclusion criterion but subsequently excluding obese participants, making consistent analysis of BMI’s impact on PE unfeasible.

Choi et al. [62] conducted a study in South Korea that included 245 pregnant women with a wide range of ages (24–43.9 years) and BMI values (16–29.6 kg/m^2^). Their control group consisted of 300 men and 227 non-pregnant women ranging in age from 25 to 82 years old, which weakens the reliability of the comparisons. The study used a reference range of 55–150 μg/dL for serum Cu levels. Among the 245 pregnant participants, only five were diagnosed with PE, with two showing elevated and three showing lower-than-normal Cu levels, though exact values were not provided. The discussion section was challenging to follow, as it primarily focused on seasonal climate patterns without clearly linking these factors to micronutrient levels in PE. The inclusion of both non-pregnant women and men as controls complicates the interpretation, raising doubts about the validity of the conclusions, particularly for data on Zn and Se.

Basu et al. [61] presented their findings only through graphical data without providing exact values. The study was conducted in women with type 1 diabetes mellitus (DM), lacking normal pregnancy data for comparison. Sarwar et al. [54] provided clinical data for pregnancies with PE, but only blood pressure measurements were shown, with no other clinical information to distinguish PE from gestational hypertension. Sak et al. [40] also omitted these critical data.

Mistry et al. [60] conducted a study on Cu, Zn, and Se and antioxidant enzymes/genes in the first 15 weeks of gestation, drawing from multicenter samples in New Zealand, Australia, and the United Kingdom. However, they did not account for potential differences among these populations within the control and PE groups, a critical factor, as genetic and lifestyle variations across these regions could significantly affect outcomes. Despite these limitations, the study found Cu and Zn levels within normal ranges. The authors proposed that the observed reduction in Se concentration and GPx activity, as well as the increased oxidative stress at delivery, might be consequences of, rather than contributors to, the oxidative stress seen in PE; thus, these findings do not support a preventative role for micronutrient supplementation, given that PE is likely to originate early in pregnancy. Furthermore, the study did not track micronutrient levels later in pregnancy (after the 20th week), missing temporal data.

The study by Rezende et al. [67] aimed to explore correlations between antiangiogenic factors and trace elements in hypertensive disorders of pregnancy. However, it lacked data on patient BMI and the timing of sample collection. The Cu levels reported were aligned with values typically seen in non-pregnant women, and the increased levels observed were consistent with those found in the second trimester of pregnancy. Without specifying the collection timing, it is not possible to verify the authors’ conclusion regarding elevated Cu levels.

Harma et al. [43] measured Zn concentrations relative to plasma protein, a non-standard method that was not clarified in the study’s methodology. Without reference values for this approach, interpreting the results as normal or abnormal is impossible. Similarly, Brito et al. [32] did not resolve whether Zn concentrations in pregnant women play a role in promoting or preventing PE, which marks the study as inconclusive. Although the report’s abstract mentions a wide age range (17–44 years) for the participants, this information was not reflected in the tables. Zn levels in pregnant women prior to delivery were below normal reference values, returning to normal post-delivery. This fluctuation suggests that the lower Zn levels observed in women with PE may be biased.

Kiran et al. [46] reported that Cu levels in PE women were within the normal reference range for non-pregnant women. However, normotensive pregnant women exhibited Cu levels (108 μg/dL) above the reference data (91 μg/dL), suggesting an elevation in Cu concentrations during pregnancy that warrants further investigation.

In contrast, Keshavarz et al. [52] and Jamal et al. [55] did not clarify whether their participants were using micronutrient supplementation, which complicates the interpretation of their findings. Both studies reported unusually high Cu levels for normal pregnancies compared to reference values. Keshavarz et al. [52] found Cu levels of 0.87 ± 0.30 mg/L (870 ± 300 μg/dL), while Abo-Elmatty et al. [39] reported 0.39 ± 0.02 mg/dL (390 ± 20 μg/dL), both far exceeding the reference range of 130–140 μg/dL for serum Cu. For PE women, Elmugabil et al. [47] observed Cu levels of 2.01 ± 0.83 mg/L, while Abo-Elmatty [53] reported 0.15 ± 0.7 mg/dL, both within the normal range for the third trimester of pregnancy. These discrepancies suggest potential misunderstandings, including the lack of information on supplementation, which could explain the elevated Cu levels in the normal pregnancy group.

Yusrawati et al. [35] conducted a study where pregnant women completed questionnaires about their nutrition, following written informed consent. Participants provided data on their nutritional status and pre-pregnancy BMI after delivery, with dietary information assessed using a Food Frequency Questionnaire (FFQ) and analyzed using the Nutrisurvey program. The study aimed to investigate the association between obesity and PE. However, the exclusive reliance on self-reported data introduces potential bias, and the study does not specify how nutrient concentrations were measured, which limits the assessment of the data’s accuracy and reliability.

The work of Gul et al. [56] uniquely reports findings that correlate Zn and Cu levels with established reference values, investigating their relationship. The Zn levels were found to be within the normal range, whereas Cu levels were elevated. However, as the study was conducted between the 28th and 40th weeks of gestation, it remains unclear how many participants were in the 28th week, a period when Cu levels may still reflect second-trimester values. Clarifying this distinction would strengthen the interpretation of the findings.

The reports investigating the role of serum Se concentrations in PE have been extensive, but they exhibit varying levels of quality and bias, as summarized in Table 3. Many studies, including those by Rayman et al. [28], Farzin et al. [65], Katz et al. [63], Negi et al. [29], Haque et al. [42], Da Silva et al. [21], Eze et al. [26], and Enebe [27], failed to properly stratify their patient populations or define the methods used to measure blood pressure in obese pregnant women, leaving significant gaps in the evaluation and diminishing the quality of their findings. This lack of rigor contributes to considerable bias in the interpretation of the results.

Other studies demonstrate even more profound issues with bias and study quality. Basu et al. [61] reported no significant differences between their control and PE groups but failed to provide statistical analyses, making their conclusions questionable. Similarly, Mistry et al. [60] claimed no significant differences between groups but used PE risk as an exclusion criterion, potentially masking any true effects.

Kim et al. [57] reported lower serum Se concentrations in women who developed PE compared to those with healthy pregnancies, a finding consistent with other studies in this analysis. However, this pattern of lower micronutrient concentrations in pre-eclamptic women is pervasive across many of the reviewed studies, raising questions about the consistency and accuracy of the data. Ghaemi et al. [25] reported significant differences between groups, but inconsistencies regarding the number of participants—mentioning 650 but only analyzing 75—raise concerns about the study’s reliability. Maleki et al. [59] found significant differences but did not appropriately stratify subjects on supplementation regimens, undermining the credibility of their results. Mistry et al. [60] reported significant differences but limited their study to a homogenous group of Caucasian participants, reducing the variability necessary for generalizable conclusions.

This trend suggests that while there are correlations between micronutrient levels and PE, the quality of the evidence is compromised by biases and methodological shortcomings. These findings underscore the need for more rigorous, high-quality studies to further investigate the role of micronutrients in the development of PE and to clarify the potential mechanisms involved.

Maleki et al. [47] found significant differences between their groups, but several of their participants were on supplementation treatments, and they did not adequately stratify the type of supplements being consumed, which undermines the credibility of their results. Mistry et al. [48] also found significant differences between groups but failed to properly stratify patients’ characteristics, further limiting the study’s validity.

Unfortunately, the insufficient data in the analyzed articles and the scattered presentation of data make a proper meta-analysis impossible. The lack of data, improper stratification, and the absence of credibility in the presented data contribute to a chaotic analysis that ultimately does not yield credible results. This reveals heterogeneity among most articles, preventing confirmation of the hypothesis.

This pattern of methodological flaws and inconsistencies is evident across most studies reviewed in this analysis, highlighting the need for more rigorous and high-quality research. The current body of evidence calls into question the veracity and coherence of the findings, emphasizing the necessity for better-controlled studies to explore the potential connection between micronutrients and the development of PE. This systematic review revealed that, based on the available data, it would not be feasible to conduct a meta-analysis, as the resulting meta-analysis would also carry bias and would not offer scientific value [71].

In 2024, Huang and Lu [72] performed a mendelian randomization analysis of genetically predicted circulating levels of beta-carotene, vitamin B6, vitamin B12, calcium, Zn, Se, Cu, folate, and phosphorus to investigate potential causal links between the risk of hypertensive disorders during pregnancy. Their findings also did not provide robust evidence to establish a causal relationship between these micronutrients and hypertensive conditions in pregnancy. In 2025, a review conducted by Ushida et al. [13] reinforced no benefits in Zn supplementation for PE prevention.

## 4. Conclusions

In conclusion, this review highlights a critical deficiency in the quality of studies investigating the relationship between micronutrient levels and the development of PE. Despite five studies on Cu being classified as high quality, their findings remain inconclusive, with three reporting no differences in Cu concentrations between normal pregnancies and PE cases, while two observed elevated levels in PE women. Regarding Zn, the only two high-quality studies available found no significant differences in Zn levels between normal and PE pregnancies. For Se, no studies met the criteria for high quality. The findings indicate that a substantial number of studies suffer from methodological limitations and potential biases, compromising the reliability of the reported results. Therefore, we emphasize the need for rigorous study design, standardized methodologies, and critical evaluation before publication to enhance the robustness and validity of research in this field.

### Study Limitations

The major limitation of our study is the language restriction, as only articles published in English or Portuguese were included, which may have led to the exclusion of relevant studies published in other languages. Another limitation of this study was the difficulty in finding normal values of micronutrients in pregnant women according to country or ethnicity. The study relied solely on the reference by Abbassi-Ghanabati et al. [69] as clinical values for comparison with the data from the reports, which may introduce bias into our results. Additionally, the high number of biased and low-quality studies hindered the performance of meta-analysis, as it would have resulted in skewed outcomes, limiting our study. Less than 10% of the manuscripts were suitable for analysis, and none included new techniques for PE detection. Furthermore, many of the studies analyzed presented biases due to the absence of a standardized protocol that considers patient age, micronutrient levels, supplementation, BMI, proteinuria, blood pressure, and other factors as inclusion and exclusion criteria. Variations in sample collection periods, materials used, and techniques applied for micronutrient detection further contributed to inconsistencies.

## Figures and Tables

**Figure 1 nutrients-17-01038-f001:**
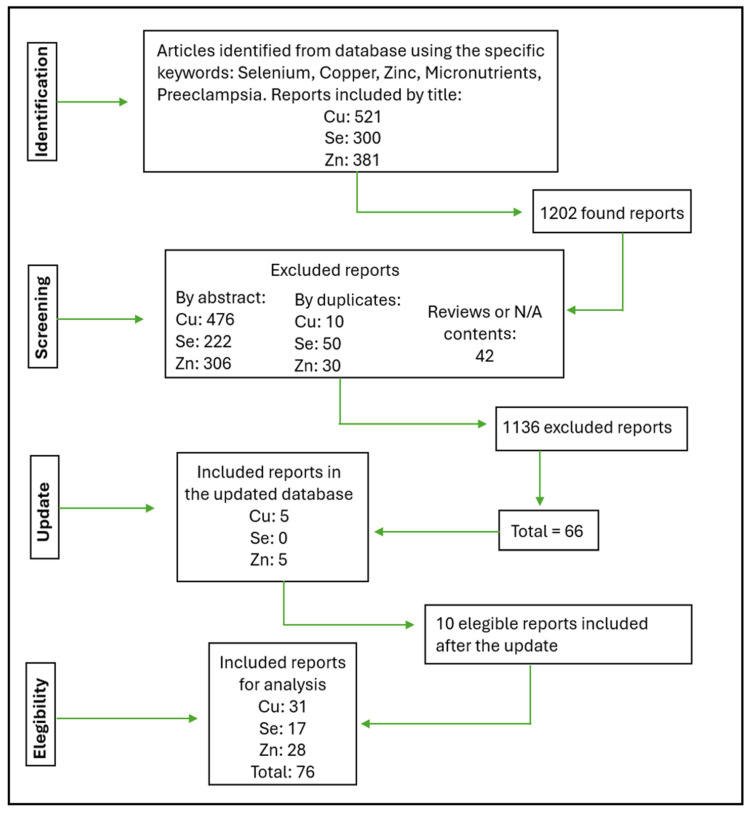
Flowchart describing how the eligibility of the articles was decided.

**Table 1 nutrients-17-01038-t001:** Reports and related micronutrients included in this review.

Micronutrient (*n* Manuscripts)	References
Zn (4)	[32,33,34,35]
Cu (6)	[36,37,38,39,40,41]
Se (6)	[21,24,25,26,28,42]
Zn and Cu (15)	[29,43,44,45,46,47,48,49,50,51,52,53,54,55,56]
Zn and Se (2)	[57,58]
Cu and Se (3)	[59,60,61]
Cu, Zn, and Se (6)	[27,29,62,63,64,65]

**Table 2 nutrients-17-01038-t002:** Analysis of quality/bias of Cu reports.

Reference	Study Type	Patients (*n*)	Age Group	Time of Measurement	Comorbidities	Outcomes	Quality	Risk of Bias
[49]	Case–control study	30 PN21 PE	PN and PE:19–31median 25	31st to 38th weeks of gestation	No comorbidities described in the study, besideshypertension	Cu (μg/dL)PN: 158.15 ± 27.86PE: 176.73 ± 25.91		
[64]	Cross-sectional prospective study	28 PN32 PE	PN:25.85 ± 3.36PE:27.00 ± 3.89	PN:36.53 ± 3.15 weeksPE:35.68 ± 2.94weeks	No comorbidities described in the study, besides hypertension	Cu (mg/L)PN: 1.458 ± 0.396PE: 2.612 ± 0.472that corresponds toCu (μg/dL)PN: 1–45.8 ± 39.6PE: 261.2 ± 47.2	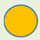	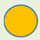
[36]	Cohort study	30 PN30 Mild PE30 Severe PE	PN:27 ± 4.1Mild PE:25 ± 2.3Severe PE:26 ± 1.9	31st to 38th weeks of pregnancy.	No comorbidities described in the study, besides hypertension	Cu (μg/dL)PN: 159 ± 38Mild PE: 188 ± 48Severe PE: 192 ± 52		
[44]	Cohort study	20 PN17 PE	PN:26.4 ± 2.7PE:24.9 ± 2.5	After the 35th week of gestation	No comorbidities were reported, besides hypertension.	Cu (µg/g placental tissue):PN: 281.0 ± 88.4PE: 269.7 ± 86.7		
[66]	Case–control study	48 PN47 PE	PN:27.92 ± 4.25PE:27.91 ± 5.21	Venous blood samples collected during weeks 29th to 38th	No comorbidities described in the study, besides hypertension	sample from placentaCu (μg/dL):PN: 2.08 ± 0.46PE: 2.65 ± 0.76		
[65]	Case–control study	60 PN60 PE	PN:26.66 ± 3.72PE:27.43 ± 3.91	30th–40th week of gestation	No comorbidities described in the study, besides hypertension and obesity	Cu (μg/dL):PN: 116.55 +1 5.23PE: 118.28 + 16.92	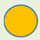	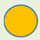
[63]	Prospective case–control study	80 PN43 PE	PN: 30.28 ± 5.7PE: 27.23 ± 7.0	PN:38.15 ± 2.19 until delivery.PE:37.72 ± 2.56 weeks until delivery.	No comorbidities described in the study	Cu (μg/dL)(1) cesarean deliverySevere PEMaternal–venous: 229.87 ± 95.59Cord–artery: 69.46± 61.63Cord–vein: 71.76 ± 58.26PN:Maternal–venous: 0.03817 ± 0.06319Cord–artery: 90.46 ± 77.32Cord–vein: 62.48± 65.82(2) vaginal delivery57.48Cord–artery: 53.93 ± 22.1Cord–vein: 52.63 ± 21.06PNMaternal–venous: 104.97± 87.74Cord–artery: 99.35 ± 81.15Cord–vein: 110.92 ± 88.54(3) maternal and fetal blood.Severe PEMaternal–venous:226.46 ± 75.17Cord–artery:58.16 ± 36.74PN:Maternal–venous:104.8 ± 85.11Cord–artery:94.9 ± 78.88Cord–vein:86.69 ± 81.26		
[29]	Case–control study	18 PN19 PE14 E	PN: 25 ± 4PE: 25 ± 4E: 28 ± 4	During the partum	No comorbidities described in the study, besides hypertension	umbilical cordCu (µg/dL)PN: 42.47 ± 8.23PE: 34.93 ± 9.53E: 26.09 ± 7.21	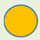	
[41]	Prospective case–control	56 PE + PN	PE + PN:31.0 ± 4.9	1st, 2nd, 3rd trimesters	No other comorbidities described in the study, besides hypertension4% had anemia, 4% DM, and smoking condition	Cu (μg/dL):PN:106 ± 12 (1st)107 ± 22 (2nd)112 ± 14.5 (3rd)PE:102 ± 8 (1st)88 ± 1 (2nd)81 ± 1 (3rd)		
[53]	Case–control study	60 PN60 PE	Not informed	N/A	No comorbidities described in the study	Cu (mg/L)The study does not show the results, just a graphicPN: 4.5PE: 3.0that corresponds toCu (ug/dL)PN: 0.45PE: 0.30		
[45]	Prospective study	37 PN30 PH30 PE	PN: 30.8PH: 30.8PE: 31.2	Gestational age at baseline, in days:PN: 261.92PH: 261.57PE: 255.83	No comorbidities described in the study, besides hypertension and obesity	Cu (μmol/L)PE: 33.91 ± 8.19PH: 33.32 ± 5.59PN: 32.04 ± 7.33that corresponds to:Cu (ug/dL)PN: 203.60 ± 46.6PH: 211.74 ± 35.5PE: 215.48 ± 52.0	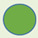	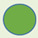
[54]	Case–control study	58 PN50 PE	PN:25.76 ± 0.73PE:25.46 ± 0.85	3rd trimester	No comorbidities described in the study, besides hypertension	Cu (mg/L)PN: 2.58 ± 0.06PE 1.98 ± 0.10that corresponds toCu (ug/dL)PN: 258 ± 6PE 198 ± 10		
[37]	Case–control study	40 PN40 HRPE40 PE	PN:31.20 ± 5.84HRPE:34.26 ± 6.69PE:31.55 ± 6.14	During the admission after the 24th week of gestation	HRPE Group:BMI ≥ 35, hypertension or PE, gestational diabetes, intrauterine growth restriction or pre-partum.	Cu (mg/L):PN: 2.014 ± 0.43PE: 1.554 ± 0.53HRPE: 1.786 + 0.51that corresponds toCu (µg/dL)PN: 201.4 ± 0.04PE: 155.4 ± 0.05HRPE: 178.6 ± 0.05		
[60]	Multi-centercase–control study	472 PN244 PE	PN: 29(23–32)PE: 28(23–32)	PN: 15th to 40th weekPE: 15th to 38th week	No comorbidities described in the study, besides hypertension and obesity	Cu (µg/L):PN: 1850.0(1663.5–2051.5)PE: 1957.4(1787.0–2177.5)that corresponds to Cu(μg/dL)PN: 185.0(166.35–205.15)PE: 195.74(178.7–217.75)		
[50]	Prospective cohort study	50 PN50 PE:(35 mild15 severe)	PN:27.10 ± 4.60PE:26.50 ± 3.90	PN31.50 ± 3.60 weeksPE:30.80 ± 3.30 weeks	No comorbidities described in the study	Cu (mg/L)PN: 1.32 ± 0.34PE:Mild: 2.26 ± 0.48Severe: 2.70 ± 0.85that corresponds toCu (μg/dL)PN: 132 ± 34Mild: 226 ± 48Severe: 270 ± 85	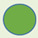	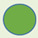
[61]	Prospective cohort study	19 PN24 PDM23 PEDM	PN:32 ± 5PDM: 30 ± 4.PEDM: 29 ± 6	3 collections:1st, 2nd, 3rd trimester	Type 1 diabetes mellitus andobesity	Higher Cu levels only at the 3rd trimester in the DM PE group.The authors do not show the levels of micronutrients in the graphics.		
[46]	Prospective cohort study	40 PN38 PE	PN: 28.7 ± 3.82PE: 29.2 ± 3.56	32–38 gestational weeks	No comorbidities described in the study	Cu (μg/dL)PN:152.45 (187.32–104.56)PE:199.5 (281.86–114.4)		
[67]	Multi-centriccase-control study	51 PN61 PH37 PE	PN: 25 ± 5PH: 27 ± 6PE: 27 ± 7	12th, 21st, and 31st gestational weeks	No comorbidities described in the study, besides hypertension	Cu (µg/dL)PN 192.0 ± 40.0PH: 196.0 ± 35.0PE: 199.0 ± 38.0	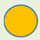	
[38]	Case–control study	60 PN70 PE	PN:29.4 ± 3.48PE:28.92 ± 4.71	28th to 36th weeks of gestation	No comorbidities described in the study, besides hypertension and obesity	Cu (μg/dL)PN: 121.21 ± 25.08PE: 210.95 ± 56.58	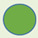	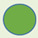
[51]	Case–control study	48 PN54 PE	PN:29 ± 5.35PE:27 ± 7.02	after the 20th week of gestation	No comorbidities described in the study	Cu (µmol/L):PN: 8.14 ± 1.80PE: 16.62 ± 3.17that corresponds toCu (μg/dL):PN: 51.47 ± 11.44PE: 105.49 ± 20.33		
[62]	Prospective cohort study	222 PN5 PE	32.0(24.0–43.9)	N/A	BMI was obtained from the self-reported weight and height	In a pooled analysis, pregnancy and neonatal outcomes when trimesters were included in statistical analysis, the prevalence of PE was significantly lower, with highCu (*p* = 0.03)median Cu (µg/dL) of all 245 pregnant women was165.0 (144.0–187.0)Cu measurement of PE was not included		
[47]	Case–control study	50 PN 50 PE	PN: 28.6 (6.6)PE: 28.6 (6.4)	PN: 36.8 weeksPE: 37.1 weeks	No comorbidities described in the study, besides hypertension and obesity	Cu (μg/dL):PN: 103.6 (86.6–126.7)PE: 111.6 (94.3–135.3)	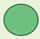	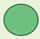
[39]	Triple-blind randomized clinical trial(interventional study)	107 PN not supplemented124 PN supplemented (1000 mg Cu oral)	18 to 35	After the 16th week up to delivery	No comorbidities described in the study	No measurement of Cu concentrations was reportedOverall incidence of PROM (7.6%), the risk of PROM wasPN: 6.7%PN supplemented: 9.9%Overall incidence of PROM (1.3%),the risk of PROM wasPN: 0.7%PN supplemented: 1.8%Incidence of PE:PN: 0%PN supplemented: 1.6%—2 casesIncidence of placenta abruption:PN: 0%PN supplemented: 0.7%—1 case	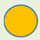	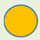
[52]	Case–control study	100 PN100 PE	PN:28.36 ± 3.26PE:28.83 ± 5.94	2nd and 3rd trimestersGestational age:PN:35.57± 2.4PE:35.13 ± 3.24	No comorbidities described in the study, besideshypertension and obesity	Cu (mg/L):PN: 2.76 ± 1.02PE: 2.01 ± 0.83that corresponds toCu (μg/dL):PN: 276 ± 102PE: 201± 83	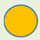	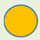
[40]	Case–control study	58 PN121 PH71 mild PE 26 severe PE 24 HELLP	PN:36.6 ± 6.6Mild PE:35.7 ± 7.28 Severe PE:36.6 ± 7.72HELLP:35.6 ± 7.54	after the 20th week of gestation	No comorbidities described in the study	PE severity related to Cu and ceruloplasmin concentration.Cu (μg/dL):PN: 62.6 ± 25.63Mild PE: 81.2 ± 11.4Severe PE: 160. 2 ± 20.89HELLP: 209. 9 ± 28.55Ceruloplasmin (mg/dL)Control: 28.3 ± 10.9Mild PE: 33.0 ± 4.81Severe PE: 65.3 ± 9.17HELLP: 92.7 ± 16.21	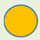	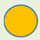
[27]	Cross-sectional analytical study	81 PN81 PE	PN:29.31 ± 5.22PE:29.53 ± 5.38	Not stated	No comorbidities described in the study, besides hypertension and obesity	Cu (mg/L)PN: 1.670 ± 1.47PE: 0.844 ± 0.57that corresponds toPN: 167 ± 147PE: 84.4 ± 57.0	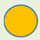	
[48]	Case–control study	60 PN60 PE	PN:26.85 ± 0.53PE:27.72 ± 0.66	The time of collection not stated, onlythird trimester	No comorbidities described in the study, besides hypertension and overweight	Cu (µg/dL)PN: 143.153 ± 3.316PE: 209.657± 8.679	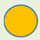	
[43]	Cohort study	44 PN24 PE	PN: 27PE: 26	Prior to or during the latent phase of labor.	No comorbidities described in the study.	Cu (µg/protein (g))PE: 47.90 ± 19.75PN: 31.60 ± 11.74Data without protein reference		
[56]	Case–control study	45 PN43 PE	PN: 27.2 ± 5.3PE: 30.2 ± 6.8	28th to 40th weeks of pregnancy	No comorbidities described in the study, besides hypertension	Cu (μg/dL)PN: 195.2 ± 38PE: 224.5 ± 26.6	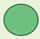	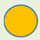
[55]	Cross-sectional and comparative study	40 PN40 PE	PN:25.46 ± 0.85PE:25.76 ± 0.73	PN:36.8 ± 0.27 weeksPE:35.32 ± 0.37 weeks	No comorbidities described in the study, besides hypertension	Cu (mg/dL)PN 0.39 ± 0.02PE: 0.15 ± 0.07that corresponds toPN: 390 ± 2.0PE: 150 ± 7.0	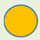	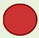

P: pregnant; PN: pregnant normotensive; PE: pregnant pre-eclampsia; PH: hypertensive pregnant; PROM: premature rupture of membranes; BMI: body mass index; Cu: copper. E: eclampsia; HRPE: high-risk pre-eclampsia; PDM: pregnant diabetes mellitus; PEDM: pregnant pre-eclampsia diabetes mellitus; HELLP: hemolysis enzymes liver low platelet syndrome.

**Table 3 nutrients-17-01038-t003:** Analysis of quality/bias of Zn reports.

Reference	StudyType	Patients (*n*)	Age Group	Time of Measurement	Comorbidities	Outcomes	**Quality**	**Risk of Bias**
[43]	Cohort study	44 PN24 PE	PN: 27.2 ± 2.7 PE: 26.7 ± 2.6	Prior to or during the latent phase of labor.	No comorbidities were described in the study.	Zn (µg/g protein) PE: 15.53 ± 4.92 PN: 11.93 ± 3.11	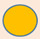	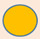
[44]	Prospective cohort study	20 PN17 PE	PN:26.4 ± 2.7 PE: 24.9 ± 2.5	After the 35th week of gestation	No comorbidities were reported, besideshypertension.	sample: placenta tissueZn (μg/g tissue):PN: 91.1 ± 36.8PE: 60.3 ± 27.8	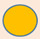	
[45]	Prospective study	30 PN30 PH30 PE	PN: 30.8 PH: 30.8 PE: 31.2	Gestational age at baseline, in days: PE: 255.83 PH: 261.57 PN: 261.92	No comorbidities were reported, besideshypertension.	Zn (μmol/L) PN: 8.85 ± 1.43 PH: 9.16 ± 1.27 PE: 9.23 ± 1.43that corresponds toZn (μg/dL)PN: 57.9 ± 9.4PH: 59.9 ± 8.3PE: 60.4 ± 9.4	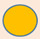	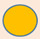
[32]	Case–control study	50 PN 20 PE 24 severePE	17–45	Two collections, 1st before the delivery, 2nd 30–45 days after partum	No comorbidities were described in the study, besides hypertension	PN: 48.3 ± 8.3 Mild PE: 50.0 ± 9.4 Severe PE: 38.8 ± 8.2		
[46]	Prospective cohort study	40 PN38 PE	PN:28.7 ± 3.82PE: 29.2 ± 3.56	32–38 gestational weeks	No comorbidities were described in the study, besides hypertension	Zn (μg/dL) PN: 108.45 (138.55–88.12) PE: 81.24 (110.65–62.4)		
[47]	Case–control study	50 PN 50 PE	PN: 28.6 (6.6)PE: 28.6 (6.4)	PN: 36.8 weeksPE: 37.1 weeks	No comorbidities were described in the study, besides hypertension and obesity	Zn (μg/dL)PN: 108.0 (91.6–131.7) PE: 102.0 (82.8–124.0)	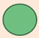	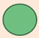
[48]	Case–control study	60 PN60 PE	PN:26.85 ± 0.53PE:27.72 ± 0.66	3rd trimester	No comorbidities were described in the study, besides hypertension and overweight	Zn (µg/dL)PN: 87.535 ± 3.710PE: 57.283 ± 1.740	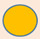	
[33]	Cross-sectional study	16 PN23 PE	PN:28.0 ± 6.9PE:31.7 ± 5.8	PN: 38(33–41)PE: 34(21–41)	No comorbidities were reported, besideshypertension.	Zn (no unit reported)PN: 43.3 (26–27)Severe PE: 47 (28–97)	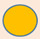	
[27]	Cross-sectional analytical study	81 PN81 PE	PN:29.53 ± 5.38PE: 29.31 ± 5.22	N/A	No comorbidities were reported, besideshypertension and obesity	Zn (mg/L)PN: 0.535 ± 0.8PE: 0.408 ± 0.39that corresponds to:Zn (µg/dL)PN: 53.5 ± 80.0PE: 40.8 ± 39.0		
[49]	Case–control study	30 PN21 PE	PN and PE: 19–31	31st to 38th weeks of gestation	No comorbidities were reported, besideshypertension	Zn (µg/dL)PN: 125.19 ± 24.23PE: 82.94 ± 28.93		
[50]	Prospective cohort study	50 PE50 PN	PN:27.10 ± 4.60 PE:26.50 ± 3.90	PN:31.50 ± 3.60 weeks PE: 30.80 ± 3.30 weeks.	No comorbidities were reported, besideshypertension.	Zn (mg/L): PN: 0.73 ± 0.33 PE: 0.71 ± 0.26 that corresponds to Zn (µg/dL)PN: 73.0 ± 33.0PE: 71.0 ± 26.0	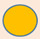	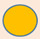
[51]	Case–control study	48 PN54 PE	PN: 29 ± 5.35 PE: 27 ± 7.02	After 20 weeks of gestation	No comorbidities described in the study.	Zn (µmol/L): PN: 8.27 ± 0.6 PE: 12.26 ± 1.83 that corresponds toZn (µg/dL)PN: 54.1 ± 3.9PE: 80.2 ± 12.0		
[52]	Case–control study	100 PN 100 PE	PN: 28.36 ± 3.26 PE: 28.83 ± 5.94	PN: 35.57 ± 2.4 PE: 35.13 ± 3.24.	No comorbidities described in the study, besideshypertension andobesity	Zn (mg/L): PN: 0.87 ± 0.30PE: 0.69 ± 0.21that corresponds toZn (µg/dL) PN: 87.0 ± 30.0PE: 69.0 ± 21.0	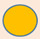	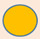
[53]	Case–control study	60 PN60 PE	N/A	After the 20th week of gestation	No comorbidities described in the study.	The results are shown only in a graph		
[57]	Cohort study	30 PN29 PE	PN: 32 PE: 32	During labor	No comorbidities described in the study.	Zn (mg/L)PN: 1.9 ± 0.5 PE: 0.7 ± 0.2 that corresponds to Zn (µg/dL)PN: 190.0 ± 50.0PE: 70.0 ± 20.0		
[54]	Case–control study	58 PN50 PE	PN: 25.76 ± 0.73PE: 25.46 ± 0.85	3rd trimester	No comorbidities described in the study.	Zn (mg/L):PN: 0.98 ± 0.03 PE: 0.77± 0.05 that corresponds toZn (µg/dL)PN: 98.0 ± 3.0PE: 77.0 ± 5.0		
[62]	Prospective cohort study	222 PN 5 PE	32.0 (24.0–43.9)	N/A	BMI was calculated from self-reported weight and height	The relationship between PE and Zn was not stated.		
[63]	Prospective case–control	80 PN43 severe PE	PN:30.28 ± 5.71Severe PE: 27.23 ± 7.09.	PN: 38.15 ± 2.19 until delivery. Severe PE: 37.72 ± 2.56 weeks until delivery.	No comorbidities described in the study.	Maternal–venous Zn (µg//L) PN: 575.5 ± 215.2 Severe PE: 652.7± 668.6 that corresponds toZn (µg/dL)PN: 57.55 ± 21.52 PE:65.27 ± 66.86		
[64]	Cross-sectional prospective study	28 PN 32 PE	PN: 25.85 ± 3.36 PE: 27.00 ± 3.89	PN: 36.53± 3.15 PE: 35.68 ± 2.94	No comorbidities described in the study, besides hypertension.	Zn (mg/L)PN: 1.086 ± 0.199 PE: 0.792 ± 0.180 that corresponds toZn (µg/dL)PN: 108.6 ± 19.9 PE: 79.2 ± 18.0	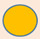	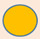
[65]	Case–control study	60 PN60 PE	PN: 26.66 ± 3.72 PE: 27.43 ± 3.91.	30th–40th weeks	No comorbidities described in the study, besides hypertension and obesity	Zn (µg/dL) PN 100.61 ± 20.12 PE: 76.49 ± 17.62	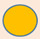	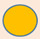
[66]	Case–control study	48 PN47 PE	PN:27.92 ± 4.25 PE:27.91 ± 5.21	29th–38th weeks.	No comorbidities described in the study.	Zn (µg/dL)PE: 1.06 ± 0.44 PN: 1.27 ± 0.41		
[29]	Case–control study	18 PN19 PE14 E	PN: 25 ± 4PE: 25 ± 4E: 28 ± 4	During the partum	No comorbidities described in the study, besides hypertension	Zn (µg/dL) PN: 83.03 ± 8.85PE: 74.43 ± 14.99E: 67.66 ± 11.71	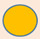	
[55]	Cross-sectional and comparative study	40 PN 40 PE	PN: 25.46 ± 0.85 PE: 25.76 ± 0.73	PN: 36.8 ± 0.27 weeks PE: 35.32 ± 0.37 weeks;	No comorbidities described in the study, besides hypertension	Zn (mg/dL)PN: 5.11 ± 0.21PE: 2.94 ± 0.45 that corresponds toZn (µg/dL)PN 5.110 ± 210PE: 2.940 ± 450	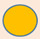	
[58]	Case–control study	23 PN 43 PE	PN: 24 ± 5 PE: 25 ± 5	PN: 37 ± 5 PE: 33 ± 5	No comorbidities described in the study, besides obesity.	Zn (mg/L): PN: 2.13 ± 3.01 PE: 18.03 ± 30.28 that corresponds toZn (µg/dL)PN: 213.0 ± 301.0PE: 1803.0 ± 3028.0		
[35]	Case–control study	70 PE70 PN	N/A	after the 20th week	No comorbidities described in the study.	Results (Zn mg) assessed by Food Frequency Questionnaire PN:5.5 (2.4–16.5)PE: 4.8 (2.39–12.87)		
[56]	Case–control study	45 PN43 PE	PN: 27.2 ± 5.3PE:30.2 ± 6.8	28th to 40th weeks of pregnancy	No comorbidities described in the study, besides hypertension	Zn (μg/dL)PN: 80.3 PE: 75.1	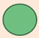	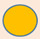
[34]	Case–control study	48 PN48 PE	20–30 years old	Third trimester	No comorbidities described in the study.	Zn (mg/dL)PN: 47.83 ± 12.72PE: 43.66 ± 11.98that corresponds to Zn (μg/dL)PN: 47,830.0 ± 12,720.0PE: 43,660.0 ± 11,980.0		

PN: Pregnant Normotensive; PE: Pregnant Pre-eclampsia; PH: Hypertensive Pregnant; E: Eclampsia; BMI: Body Mass Index; Zn: Zinc.

**Table 4 nutrients-17-01038-t004:** Analysis of quality/bias of Se reports.

Reference	Study Type	Patients (*n*)	Age Group	Time of Measurement	Comorbidities	Outcomes	Quality	Risk of Bias
[28]	Case–control study	53 PN53 PE	PN: 30 ± 5PE: 31 ± 5	Toenail sample collected 3–12 months before	No comorbidities were reported, besides hypertension.	Toenail sample: Se (mg/Kg)PN: 0.62 (0.57–0.69) PE: 0.56 (0.51–0.64)		
[60]	Cross-sectional cohort study	27 PN25 PE	PN: 29 ± 6PE: 32 ± 5	PN: 37 ± 5PE: 33 ± 5	No comorbidities described in the study, besides hypertension and obesity	Se (μg/L)PN: 58.4 ± 14.9 PE: 39.7 ± 13.8 that corresponds to Se (μg/dL)PN: 5.84 ± 1.49 PE: 3.97 ± 1.38		
[58]	Case–control study	23 PN43 PE	PN: 24 ± 5PE: 25 ± 5	N/A	No comorbidities described in the study, besides obesity.	Hair sampleSe (μg/g)PN: 23.93 ± 2.62 PE: 24.42 ± 1.78 Se serum (mg/L)PN: 0.14 ± 0.01PE: 0.06 ± 0.01 that corresponds toSe (μg/dL)PN: 14.0 ± 1.0PE: 6.0 ± 1.0		
[59]	Case–control study	40 PN40 PE	PN:26.42 ± 3.73PE:27.62 ± 5.25	34th to 39th weeks	No comorbidities described in the study.	Se (μg/L)PN: 58.51 ± 11.85 PE: 51.75 ± 11.62 that corresponds toSe (μg/dL)PN: 5.851 ± 1.185 PE: 5.175 ± 1.162		
[64]	Cross-sectional prospective study	28 PN32 PE	PN: 25.85 ± 3.36PE: 27.00 ± 3.89	PN:36.53 ± 3.15PE:35.68 ± 2.94	No comorbidities described in the study, besides hypertension.	Se (mg/L)PN: 87.50 ± 10.96 PE: 60.68 ± 6.42that corresponds toSe (μg/dL)PN: 8750.0 ± 1096.0 PE: 6068.0 ± 642.0	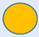	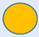
[29]	Case–control study	18 PN19 PE 14 E	PN: 25 ± 4PE: 28 ± 4E: 28 ± 4	During the partum	No comorbidities described in the study, besides hypertension	Se (µg/dL) PN: 22.17 ± 4.19 PE: 18.58 ± 5.21 E: 16.34 ± 5.23	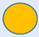	
[63]	Case–control study	80 PH43 HRPE	PN: 30.28 ± 5.7PE:27.23 ± 7	Before the partum	No comorbidities described in the study.	Se (µg/L) from the placenta after birthPN: 110.7 ± 19.4Severe PE: 98.26 ± 24.2 that corresponds toPN: 11.07 ± 1.94Severe PE: 9.826 ± 2.42		
[57]	Case–control study	30 PN29 PE	PN: 31 ± 3.1PE: 32.1 ± 4.6	PN:39.1± 1.1PE:34.1 ± 3.0	No comorbidities described in the study.	Se (mg/L)PN: 0.09 ± 0.01PE: 0.09 ± 0.01 that corresponds toSe (µg/L)PN:9.0 ±1.0PE: 9.0 ± 1.0		
[65]	Case–control study	60 PN60 PE	PN:26.66 ± 3.72PE: 7.43 ± 3.91	30th–40th weeks of gestation	No comorbidities described in the study, besides hypertension and obesity.	Se (µg/dL)PN: 10.47 ± 2.78 PE: 8.82 ± 2.10	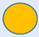	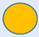
[25]	Prospective case–control study	38 PN38 PE	PN:28.2 ± 3.12PE:28.4 ± 3.13	From the 24th–28th weeks of gestation, and followed up 48 h after the partum	No comorbidities described in the study, besides overweight	Se (µg/dL)PN: 82.03 ± 15.54 PE: 70.63 ± 21.41		
[61]	Prospective cohort study	19 PN24 PDM23 PEDM	PN: 32 ± 5PDM: 30 ± 4 PEDM: 29 ± 6	3 collections:1st, 2nd, 3rd trimesters	Diabetes mellitus type 1 and obesity	Data was shown only in graphics.The serum levels of Se were not significantly different between the three analyzed groups.		
[42]	Case–control study	118 PN74 PE	PN: 24 ± 0.51PE: 27 ± 0.98	6 h apart from patients and 20 gestational weeks	No comorbidities described in the study.	Se (µg/L) rural area:PN: 32.18 ± 1.22 µg/L PE: 23.07 ± 0.96 µg/L that corresponds toSe (µg/dL)PN: 3.218 ± 0.122PE: 2.307 ± 0.096Se (µg/L) urban area:PN: 24.67 ± 0.75PE: 32.29 ± 2.09that corresponds toSe (µg/dL)PN: 2.467 ± 0.075PE: 3.229 ± 0.209	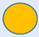	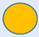
[21]	Case–control study	32 PN20 PH38 PE	PN:27.9 ± 6.4 PH ± PE:28.7 ± 6.828	During patient admission.	High level of comorbidities, including recurrent urinary tract infection, kidney stones, depression, and heart disease, besides hypertension and obesity	Se (µg/dL)PN: 56.4 ± 15.3PH: 53.2 ± 15.2PE: 53.3 ± 16.8		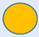
[62]	Prospective cohort study	222 PN 5 PE	median age 32.0 years (24.0–43.9)	N/A	BMI was calculated from self-reported weight and height	The relationship between PE and Se was not stated		
[27]	Cross-sectional analytical study	81 PN 81 PE	PN:29.53 ± 5.38PE:29.31± 5.22	N/A	No comorbidities were reported, besideshypertension and obesity	Se (mg/L)PN: 0.842 ± 0.71 mg/LPE: 01.758 ± 3.35 mg/Lthat corresponds toSe (µg/dL)PN: PN: 84.2 ± 71.0PE: 175.8 ± 335.0	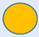	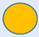
[26]	Comparative case–control study	56 PN56 PE	PN: 31.45 ± 4.9PE:31.03 ± 4.73	N/A	No comorbidities described in the study, besides obesity	Se (µmol/L)PN:1.20 ± 0.46 µmol/LPE: 0.67 ± 0.27 µmol/Lthat corresponds to Se (µg/dL)PN: 94.75 ± 36.38PE: 52.91 ± 21.32	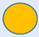	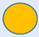
[24]	Cohort study	363 PN121 PH 12 PE	35.1 ± 4	1st trimester and 12 weeks post-partum	No comorbidities described in the study, besides hypertension, obesity, and smoking condition	Se (µg/L) PN: 63.58 (48.35–90.14)PH + PE: 57.69 (44.63–70.13)that corresponds toSe (µg/dL)PN: 6.358 (4.835–9.014)PH + PE: 5.769 (4.463–7.013)		

PN: Pregnant Normotensive; PE: Pregnant Pre-eclampsia; PH: Hypertensive Pregnant; BMI: Body Mass Index; PDM: Pregnant Diabetes Mellitus; PEDM: Pregnant Pre-eclampsia Diabetes Mellitus; HRPE: High Risk of Pre-eclampsia.

**Table 5 nutrients-17-01038-t005:** Experimental measurements of Cu, Zn, and Se references during normal pregnancy.

	Non-Pregnant	1st Trimester	2nd Trimester	3rd Trimester
**Cu**	63–158 µg/dL	112–199 µg/dL	165–221 µg/dL	130–240 µg/dL
**Zn**	75–120 µg/dL	57–88 µg/dL	51–80 µg/dL	50–70 µg/dL
**Se**	63–160 µg/L	116–146 µg/L	75–145 µg/L	71–133 µg/L

Data according to Abbassi-Ghanabati et al. [69].

## Data Availability

The original contributions presented in the study are included in the article. All other information can be requested from the corresponding author.

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
