# Peer review of "Copper, Zinc, and Selenium Levels During Pregnancy and Their Relationship with Pre-Eclampsia Risk: A Systematic Review"

_nutrients, 2025, doi:10.3390/nu17061038_

Round 1
Reviewer 1 Report
Comments and Suggestions for Authors
Manuscript ID: nutrients-3488844
Type of manuscript: Review
Title: Copper, Zinc, and Selenium Levels During Pregnancy and Their Relationship to Preeclampsia Risk: A Systematic Review.
The present study aims to perform a systematic review according to PRISMA guideline, using Cochrane risk of bias tool and the quality of the report and Joanna Briggs Critical Appraisal Tools to answer the question if the deficiency of zinc, copper and selenium in pregnant women might develop or induce preeclampsia (PE), to support clinical and governmental health care actions.
Comments and Suggestions for Authors:
The manuscript is an interesting review but requires some consideration.
An important point is that if the inclusion criterion is "pregnant woman", it is not understood how the results include data from studies of non-pregnant women and men. Likewise, the inclusion criterion "any age" causes data from patients with an age outside the reproductive range to appear in the results. Although "the results comparing the PE woman with non-pregnant women or men, were considered red in bias and low quality also", all of this should have been removed for the systematic review because it causes it to be misinterpreted.
Other issues to consider:
After having been introduced, acronyms such as preeclampsia (PE), copper (Cu), selenium (Se), zinc (Zn), etc., continue to be used without acronyms in the manuscript or are reintroduced unnecessarily. This should be reviewed throughout the manuscript.
Page 2, Line 83. The phrase "Nowadays, the supplementation of calcium for pregnant women at the risk of PE has been recommended at doses of 1,000 to 2,000 mg/day since 2011 by WHO [2]" is a reiteration and should be deleted.
Page 5, Line 153. Where it says "eclapsia" it should say eclampsia.
Tables 2, 3 and 4 should include table footers with all the acronyms used in them.
Page 18, Line 241. Where it says "table 5" it should say Table 5. These aspects should be reviewed throughout the manuscript.
Page 20, Line 326. Where it says "Rayman" it should say Rayman et al. [24]. These aspects should be reviewed throughout the manuscript.
Page 21, Line 340. Where it says "Roberts and Schilsky et al." bibliographic reference numbers should be included. These aspects should be reviewed throughout the manuscript.
Author Response
Reviewer 1.
We thank the reviewer for the thorough revision of our work. We agree with all the points raised, which were of utmost importance in improving the quality of the study
Comment 1: An important point is that if the inclusion criterion is "pregnant woman", it is not understood how the results include data from studies of non-pregnant women and men. Likewise, the inclusion criterion "any age" causes data from patients with an age outside the reproductive range to appear in the results. Although "the results comparing the PE woman with non-pregnant women or men, were considered red in bias and low quality also", all of this should have been removed for the systematic review because it causes it to be misinterpreted.
Response1.: corrected and removed
Other issues to consider:
Comment 2. After having been introduced, acronyms such as preeclampsia (PE), copper (Cu), selenium (Se), zinc (Zn), etc., continue to be used without acronyms in the manuscript or are reintroduced unnecessarily. This should be reviewed throughout the manuscript.
Response 2: corrected
Comment 3. Page 2, Line 83. The phrase "Nowadays, the supplementation of calcium for pregnant women at the risk of PE has been recommended at doses of 1,000 to 2,000 mg/day since 2011 by WHO [2]" is a reiteration and should be deleted.
Response 3: deleted
Comment 4: Page 5, Line 153. Where it says "eclapsia" it should say eclampsia.
Response 4: Corrected
Comment 5: Tables 2, 3 and 4 should include table footers with all the acronyms used in them.
Response 5. Included
Comment 6 : Page 18, Line 241. Where it says "table 5" it should say Table 5. These aspects should be reviewed throughout the manuscript.
Response 6: corrected
Comment 7: Page 20, Line 326. Where it says "Rayman" it should say Rayman et al. [24]. These aspects should be reviewed throughout the manuscript.
Response 7: reviewed in all manuscript
Commnet 8: Page 21, Line 340. Where it says "Roberts and Schilsky et al." bibliographic reference numbers should be included. These aspects should be reviewed throughout the manuscript.
Response 8: reviewed in all manuscript

Reviewer 2 Report
Comments and Suggestions for Authors
This is an interesting systematic review article with adequate novelty. This systematic review has a methodological aspect of high quality and presentation of its results. Some points should be addressed.
- In the Introduction section, the sentence in lines 80-82 "According to the
Oswaldo Cruz Foundation (FIOCRUZ) there is no scientific evidence that the use of vitamins D, C and E to reduce the prevalence of PE" needs more analysis. - The size of Figure 1 should be increased.
- The abbreviation for preeclampsia (PE) should only be explained at the begining of the article. The authors should check their manuscript to use only the abbreviation "PE" after the first explanation.
- Again the authors should check the manuscript to use relevant abbreviation, e.g. Zn, Cu, Se, BMI, etc. after the first explanation.
- In the Discussion section the paragraph in lines 286-295 needs relevant references.
- In line 371 "Sawar provided clinical data ....", a reference should be added.
Author Response
Reviewer 2
We thank the reviewer for the thorough revision of our work. We agree with all the points raised, which were of utmost importance in improving the quality of the study.
All the correction is yellow highlighted in the manuscript
Comments:
This is an interesting systematic review article with adequate novelty. This systematic review has a methodological aspect of high quality and presentation of its results. Some points should be addressed.
Comment 1. In the Introduction section, the sentence in lines 80-82 "According to the
Oswaldo Cruz Foundation (FIOCRUZ) there is no scientific evidence that the use of vitamins D, C and E to reduce the prevalence of PE" needs more analysis.
Response 1. Yes, you are right. We remove Fiocruz and include more information (lines 89-99) and a reference
Ushida T, Tano S, Matsuo S, Fuma K, Imai K, Kajiyama H, Kotani T. Dietary supplements and prevention of preeclampsia. Hypertens Res. 2025 Feb 10. doi: 10.1038/s41440-025-02144-9.
Comment 2.The size of Figure 1 should be increased
Response 2. Increased
Comment 3. The abbreviation for preeclampsia (PE) should only be explained at the begining of the article. The authors should check their manuscript to use only the abbreviation "PE" after the first explanation.
Comment 4. Again the authors should check the manuscript to use relevant abbreviation, e.g. Zn, Cu, Se, BMI, etc. after the first explanation.
Responses 3 and 4: corrected
Comment 5. In the Discussion section the paragraph in lines 286-295 needs relevant references.
Response 5. As our work was very complex and the majority of the articles did not include clinical reference values, we chose this article (above) as the basis for the entire discussion to avoid any conflict of interest. We chose the Gynecology journal as reference, which provides a database for medical information.
Abbassi-Ghanavati, M.; Greer, L.G.; Cunningham, F.G. Pregnancy and Laboratory Studies: A Reference Table for Clinicians. Obstet. Gynecol. 2009, 114, 1326–1331,
Comment 6: In line 371 "Sawar provided clinical data ....", a reference should be added.
Response 6. included

Reviewer 3 Report
Comments and Suggestions for Authors
thank you for the opportunity to review the MS entitled Copper, Zinc, and Selenium Levels During Pregnancy and Their Relationship to Preeclampsia Risk: A Systematic Review. The review is using adequate and high-level tool to deem articles included in the review in terms of risk of bias and quality. The topic is very interesting and omitted in the literature, although the lack of these micronutrients may play a role a crucial role in the pathophysiology of preeclampsia.
The review is comprehensive and it encompasses all of the literature articles on this topic, and may attract much attention and citations from the readers. Indeed, I would like to recommend the MS for publication which is aligned with the profile of the Nutrients.
I have some issues with improving the MS, although it is not easy to publish a review on this issue due to the scarcity of good quality papers, confusing and contradictory results and inadequate study design.
First of all, the results include just data on the quality and risk of the studies. The results section should interpret some core results on PE and micronutrients. The discussion section seems to be a mixture of the conclusion and results. Almost all important arguments and results interpretation occur in this section.
The considerations is just repeating the results and the conclusions so I think it should be committed. The authors in this section mention the role of PIGF and sonographic markers in predicting PE which is beyond the scope (between lines 500 and 509).
The Conclusions section has to be more conclusive/informative: i.e. Cu levels in pregnancies complicated with PE were lower as compared to their healthy counterparts but in the normal range according to the good quality studies. So the authors should focus on the results that can be extracted from unbiased, high-quality studies.
There are some minor issues:
In the abstract: the definition of preeclampsia should be omitted (lines 19-21).
The results in the abstract section are scarcely presenting the results (lines 31-32). The authors set too much focus on the biases and the study design flows throughout the MS, but they do not emphasize the analyzable results.
Line 117: the authors marked the studies as red referring to the bias category, but they inform us about it 3 lines below.
In the results Lines 129 - 140 can be shortened. Table 1 is self-explanatory and interprets well the results. It is not necessary to repeat the tables in the text.
Lines 151-153 are explanation of the tables and should be involved in the table legend, not in the text.
Table 2 the last 2 columns regarding the quality and risk of bias are missing.
The analysis of the Table 2 results gives an overview of the papers only just in terms of quality and bias, but nothing else. The Discussion contains the results and the conclusion with various interpretations and arguments.
The beginning of the Discussion refers to the Brazilian constellation of the nutrition status of pregnant women which I don't feel is necessary (lines 171-176 and lines 180-181).
The description of the role of the Zn, Cu and Se must be in the Introduction (lines 186-220). The discussion is the explanation of the results with the relevant background information and not the pure interpretation of the essence of these micronutrients. The ionisation number of the ions has to be placed in the upper corner and not down: Se2+, Cu2+,Zn2+.
Line 263: there is decimal point in English and there is not decimal comma.
Grammatical faults: Lines 300-308 - ... [34] reported Cu levels of ... and [40] reported levels ... are not correct. Here, you need to write out the authors. In the next sentence, the reference has been omitted.
References are missing in line 371. And Sawar should be replaced by Sawar et al. (REF)
In line 382: GPx is not known what it is.
Grammatically incorrect:
Line 388 Rezende et al. [20] study aimed...
Reference is omitted in line 398 (Brito et al).
Grammatically incorrect: lines 412-414
[40] reported 0.39 ± 0.02 mg/dL (390 ± 20 μg/dL), both far exceeding the reference range of 130–140 μg/dL for serum Cu. For PE women, [34] observed Cu levels of 2.01 ± 0.83 mg/L, and [40] reported 0.15 ± 0.7 mg/dL, which were within...
Line 442: control and study group can be understandable. Both case and control groups can be study groups.
Author Response
Reviewer 3.
We thank the reviewer for the thorough revision of our work. We agree with all the points raised, which were of utmost importance in improving the quality of the study. Once again, we appreciate the reviewer's time and dedication.
General Comment
thank you for the opportunity to review the MS entitled Copper, Zinc, and Selenium Levels During Pregnancy and Their Relationship to Preeclampsia Risk: A Systematic Review. The review is using adequate and high-level tool to deem articles included in the review in terms of risk of bias and quality. The topic is very interesting and omitted in the literature, although the lack of these micronutrients may play a role a crucial role in the pathophysiology of preeclampsia.
The review is comprehensive and it encompasses all of the literature articles on this topic, and may attract much attention and citations from the readers. Indeed, I would like to recommend the MS for publication which is aligned with the profile of the Nutrients.
I have some issues with improving the MS, although it is not easy to publish a review on this issue due to the scarcity of good quality papers, confusing and contradictory results and inadequate study design.
Comments:
Comment 1. First of all, the results include just data on the quality and risk of the studies. The results section should interpret some core results on PE and micronutrients. The discussion section seems to be a mixture of the conclusion and results. Almost all important arguments and results interpretation occur in this section.
Response 1: As the work is complex and extensive, the presentation of risk of bias and quality assessment results, as well as the outcomes, are provided in the table. However, the interpretation of these PE results is included in the text. Therefore, for better understanding, instead of dividing them into two sections, we combined them into Results and Discussion.
Comment 2. The considerations is just repeating the results and the conclusions so I think it should be committed. The authors in this section mention the role of PIGF and sonographic markers in predicting PE which is beyond the scope (between lines 500 and 509).
Response 2. We agree. Removed
Comment 3. The Conclusions section has to be more conclusive/informative: i.e. Cu levels in pregnancies complicated with PE were lower as compared to their healthy counterparts but in the normal range according to the good quality studies. So the authors should focus on the results that can be extracted from unbiased, high-quality studies.
Response 3. We rewrite de conclusions
The discussion of good quality reports are shown in lines 312-324.
Comments: There are some minor issues:
Comment 4. In the abstract: the definition of preeclampsia should be omitted (lines 19-21).
Response: removed
Comment 5: The results in the abstract section are scarcely presenting the results (lines 31-32). The authors set too much focus on the biases and the study design flows throughout the MS, but they do not emphasize the analyzable results.
Response: we rewrite the abstract including this information
Comment 6: Line 117: the authors marked the studies as red referring to the bias category, but they inform us about it 3 lines below.
Response 6: we agree and corrected
Comment 7. In the results Lines 129 - 140 can be shortened. Table 1 is self-explanatory and interprets well the results. It is not necessary to repeat the tables in the text.
Response 7. We agree and remove it
Comment 8. Lines 151-153 are explanation of the tables and should be involved in the table legend, not in the text.
Response 8: corrected
Comment 9: Table 2 the last 2 columns regarding the quality and risk of bias are missing.
Response 9: reviewed. All columns are ok.
Comment 10. The analysis of the Table 2 results gives an overview of the papers only just in terms of quality and bias, but nothing else. The Discussion contains the results and the conclusion with various interpretations and arguments.
Response 10: All tables have the information about outcomes (PE/PN) results. Reviewed
Comment 11: The beginning of the Discussion refers to the Brazilian constellation of the nutrition status of pregnant women which I don't feel is necessary (lines 171-176 and lines 180-181).
Response 11: Removed
Comment 12: The description of the role of the Zn, Cu and Se must be in the Introduction (lines 186-220). The discussion is the explanation of the results with the relevant background information and not the pure interpretation of the essence of these micronutrients. The ionisation number of the ions has to be placed in the upper corner and not down: Se2+, Cu2+,Zn2+.
Response 12: We agree and changed. Highlight in introduction and results/discussion
Comment 13: Line 263: there is decimal point in English and there is not decimal comma.
Response 13: corrected
Comment 14: Grammatical faults: Lines 300-308 - ... [34] reported Cu levels of ... and [40] reported levels ... are not correct. Here, you need to write out the authors. In the next sentence, the reference has been omitted.
Response 14. Corrected
Comment 15. References are missing in line 371. And Sawar should be replaced by Sawar et al. (REF)
Response 15: corrected
Comment 16: In line 382: GPx is not known what it is.
Response 16: it is glutathione peroxidase (included)
Comment 17: Comment Grammatically incorrect:
Line 388 Rezende et al. [20] study aimed...
Response 17. Corrected
Comment 18: Reference is omitted in line 398 (Brito et al).
Response 18: Corrected
Comment 19 Grammatically incorrect: lines 412-414
[40] reported 0.39 ± 0.02 mg/dL (390 ± 20 μg/dL), both far exceeding the reference range of 130–140 μg/dL for serum Cu. For PE women, [34] observed Cu levels of 2.01 ± 0.83 mg/L, and [40] reported 0.15 ± 0.7 mg/dL, which were within...
Response 19: corrected
Comment 20: Line 442: control and study group can be understandable. Both case and control groups can be study groups.
Response 20: Corrected

Round 2
Reviewer 1 Report
Comments and Suggestions for Authors
Round 2.
In the new V2 manuscript, the authors have made changes based on the recommendation of referee that improve its presentation. However, authors should try to improve the presentation of the manuscript and make it more accessible to the reader.
Acronyms such as PE should be avoided in the title.
The results of the tables continue to be presented with confusion.
Values ​​for non-pregnant women are presented. This does not respond to the objective of the review (Cu, Zn, and Se Levels During Pregnancy and Their Relationship to PE Risk). The review should focus on the differences between normal pregnancies and those with PE.
The results would be clearer by ignoring the data on non-pregnant women, women with ages outside the reproductive range, and men.
Acronyms appear that are not included in the footer of tables such as NT, NP, GHT, HTN, E, CG. The most necessary acronyms should be selected andonly those used.
Phrases appear in Portuguese like:Valores confusos do desvio padrão, em ambos os micro-nutrientes avaliados em nossa revisão, em especial no Zno, indica um resultado enviesado, Variação muito grande, Comparou apenas com NP or Valores elevados do desvio padrão, especificamente no cobre, indica um resultado enviesado. The quality of the results presented in the tables should be carefully reviewed.
In Page 23, Line 438 there is an error in the bibliographic references.
Two Discussion sections are presented: in Page 4, Line 171 (Results and Discussion) and in Page 19, Line 232 (Discussion). This confusion should be avoided in the manuscript.
Author Response
To reviewer 1
We deeply appreciate the reviewer for their valuable considerations regarding the errors in the tables, which we had not previously noticed, and we apologize for this inconvenience. We carefully reviewed them, removed incorrect texts, and excluded the groups of non-pregnant women and men. We standardized the values so that the units of measurement for micronutrients are in µg/dL for those that were presented in different units, making it easier for the reader to make comparisons.
The reviewer comments:
In the new V2 manuscript, the authors have made changes based on the recommendation of referee that improve its presentation. However, authors should try to improve the presentation of the manuscript and make it more accessible to the reader.
We thank for the detailed revision made by reviewer that improve the work. We agree and corrected it. The table was revised. All new modifications are in magenta
Comment 1. Acronyms such as PE should be avoided in the title.
Response 1. The Acronym in the title was corrected.
Comment 2. The results of the tables continue to be presented with confusion.
Values ​​for non-pregnant women are presented. This does not respond to the objective of the review (Cu, Zn, and Se Levels During Pregnancy and Their Relationship to PE Risk). The review should focus on the differences between normal pregnancies and those with PE.
The results would be clearer by ignoring the data on non-pregnant women, women with ages outside the reproductive range, and men.
Response 2. As for the comparisons made in some articles including non-pregnant individuals and men, they were initially included to highlight the flaws in the quality of the articles. However, we agree with the reviewer that this makes the tables confusing and remove this information from the table.
Comment 3. Acronyms appear that are not included in the footer of tables such as NT, NP, GHT, HTN, E, CG. The most necessary acronyms should be selected andonly those used.
Response 3. Corrected
Comment 4. Phrases appear in Portuguese like:Valores confusos do desvio padrão, em ambos os micro-nutrientes avaliados em nossa revisão, em especial no Zno, indica um resultado enviesado, Variação muito grande, Comparou apenas com NP or Valores elevados do desvio padrão, especificamente no cobre, indica um resultado enviesado. The quality of the results presented in the tables should be carefully reviewed.
Response 4. Mistaken phrases in Portuguese were excluded. We reviewed the tables
Comment 5. In Page 23, Line 438 there is an error in the bibliographic references.
Response 5. Bibliography in line 438 corrected
Comment 6. Two Discussion sections are presented: in Page 4, Line 171 (Results and Discussion) and in Page 19, Line 232 (Discussion). This confusion should be avoided in the manuscript.
Response 6. Mistaken Discussion section in line 232 corrected/excluded.

Reviewer 3 Report
Comments and Suggestions for Authors
NA
Author Response
Thank you very much for reviewing our work.
Kind regards